# Psycho-Educational and Rehabilitative Intervention to Manage Cancer Cachexia (PRICC) for Advanced Patients and Their Caregivers: Lessons Learned from a Single-Arm Feasibility Trial

**DOI:** 10.3390/cancers15072063

**Published:** 2023-03-30

**Authors:** Loredana Buonaccorso, Stefania Fugazzaro, Cristina Autelitano, Elisabetta Bertocchi, Monia Allisen Accogli, Monica Denti, Stefania Costi, Gianfranco Martucci, Luca Braglia, Maria Chiara Bassi, Silvia Tanzi

**Affiliations:** 1Scientific Directorate, Azienda USL-IRCSS di Reggio Emilia, 42122 Reggio Emilia, Italy; 2Physical Medicine and Rehabilitation Unit, Azienda USL-IRCSS di Reggio Emilia, 42122 Reggio Emilia, Italy; 3Palliative Care Unit, Azienda USL-IRCSS di Reggio Emilia, 42122 Reggio Emilia, Italy; 4Department of Surgery, Medicine, Dentistry and Morphological Sciences, University of Modena and Reggio Emilia, Via del Pozzo No. 74, 41100 Modena, Italy; 5Local Network of Palliative Care, AUSL Modena, 41100 Modena, Italy; 6Medical Library, Azienda USL-IRCSS di Reggio Emilia, 42122 Reggio Emilia, Italy

**Keywords:** cancer cachexia, exercise, cancer rehabilitation, palliative medicine, non-pharmacologic treatment, psycho-educational treatment, psycho-social treatment

## Abstract

**Simple Summary:**

Approximately half of all patients with advanced cancer experience cachexia, and the prevalence rises above 80% in the last weeks of life. Cancer cachexia (CC) is characterised by anorexia, systemic inflammation, ongoing loss of muscle mass, and weight loss, affecting the quality of life. Different treatments are proposed to manage CC, and comprehensive treatment requires a multidisciplinary approach aimed at evaluating overall patients’ conditions and relieving the symptoms, including psycho-social and spiritual functions. Key elements in CC management are personalized, multi-targeted, and multimodal interventions, but it is hard for some patients to follow programs based on several components. We examined the feasibility of a bimodal intervention, including a psycho-educational component and exercises, to support patients and their caregivers in managing CC. Our prospective mixed-methods pilot study explored feasibility data, changes in PROMS and performance outcomes over time, and qualitative data from dyads and healthcare professionals’ interviews.

**Abstract:**

Background: Key elements in cancer cachexia (CC) management are personalized and multimodal interventions, but it is hard for some patients to follow programs based on several components. We examined the feasibility of a bimodal intervention, including a psycho-educational component and exercises, to support patients and their caregivers in managing CC; Methods: Prospective mixed-methods pilot study explored feasibility data, changes in patient-reported outcomes, and performance outcomes over time in a convenient sample of 30 consecutive CC patients and their caregivers. Results: Twenty-four dyads consented to participate. Twenty dyads received at least two psycho-educational sessions, so the psycho-educational component was feasible for 83.3% of the sample. Six dyads participated in at least fourteen out of twenty-seven rehabilitation sessions, so the exercise program was feasible for 25.0% of the sample. Six dyads showed compliance greater than 50% for both components of the bimodal intervention. Conclusions: While we did not meet our primary feasibility endpoint and had mixed acceptability, our experience provides insight into the challenges and lessons learned in implementing a primary palliative care intervention for CC. More robust studies are needed to help clinicians understand the best exercise program for CC patients, to be included in a multimodal intervention.

## 1. Introduction

Approximately half of all patients with advanced cancer experience cachexia, with the prevalence rising above 80% in the last weeks of life [1,2]. Cancer cachexia (CC) is defined as “disease-related malnutrition”, based on the Global Leadership Initiative in Malnutrition [3] and “the presence of systemic inflammation”, described as accelerated protein breakdown and the hallmark of muscle loss driven by metabolic changes [2,3]. CC is a continuum with three stages of clinical relevance: pre-cachexia, cachexia, and refractory cachexia [4]. Based on the evolution of CC, different interventions are proposed for patients. The treatments must be related to a comprehensive assessment of the clinical condition of the patients, resulting in personalized, multi-targeted, and multimodal treatments [2,5]. The components of such multimodal support may target caloric intake, physical activity, psycho-social, and spiritual functions, as well as key factors in cachexia pathophysiology, such as inflammation [1,2,5]. Evidence-based practice indicates that it is difficult for patients to comply with all of these components, with the intake of supplements, non-steroid medications, and anti-inflammatory drugs being the most abandoned components [6,7]. Especially among patients in palliative care (PC), a 20% dropout index has been detected [6,7,8,9].

Exercise can reduce the effects of CC by modulating muscle metabolism, reducing insulin resistance, and decreasing the inflammatory cascade [10,11]. Physical activity seems to act as an important anabolic stimulus, especially for patients on chemotherapy, but little is known about its efficacy in the advanced stages of the disease [2,3,4]. The most recent Cochrane review on this subject examined the safety, acceptability, and efficacy of exercise in adult patients with cachexia, including only four randomized control trials (RCT) that studied exercise programs in patients with CC [12]. The review did not consent to a quantitative analysis of data, but a qualitative synthesis was made, and the authors concluded that exercise may be helpful, but there are insufficient data to consider its effectiveness on fatigue, muscle weakness, and quality of life (QoL) [12]. Considering the characteristics of these patients, an exercise program must be tailored and feasible at home and include resistance exercises as a key component. Some studies have highlighted that resistance exercises stimulate the building of muscle mass and increase strength, while the repetition of exercise is useful in reducing the inflammatory response [5,10,11]. The authors who have studied this topic proposed an intervention with a duration between 5 and 8 weeks and moderate exercise intensity [9,12]. Other important components of the physical activity program aimed at motivating the patient and supporting compliance for goal setting [10,13] stretching and relaxation exercises [14].

Concerning CC, patients and their families experience stressful changes in eating habits and challenging social interactions [15,16]. The psycho-social effects of CC are described as “negative emotions associated with reduced dietary intake, involuntary weight loss, and social consequences from these symptoms” [15]. The reduced appetite was a cause of concern for the patients, who were further distressed when pressured to eat by their family carers [17,18]. From the family member’s point of view, feelings of guilt and anxiety were experienced when their loved one rejected the food, which was seen as a means of hope to continue to live [16,17]. Thus, changes in eating habits disrupt the relational balance and harmony within the family, explaining greater eating-related distress in patients living with their partners than in those living alone [17,18,19]. The change in weight, with a clear knowledge of cachexia, was another major source of patients’ distress. Self-image is an important dimension of personal identity and negative changes such as weight loss may lead to an altered self-perception. The wasted appearance was a barrier to social engagement because patients were worried about people’s reactions to their physical aesthetics, which reminded them of being ill [15]. Moreover, also the family carers suffered from CC in seeing their loved one waste away [19,20,21]. 

The psycho-social component of multimodal intervention for CC patients enabled stress management and coping, improved body image, QoL, treating depression, and supporting adherence to exercise, physical activity, and nutritional care [21]. Recently, as a specialized palliative care team (SPCT) in the hospital, we conducted a scoping review on non-pharmacological intervention for CC in advanced patients [22]. Data showed two important topics: (1) most of the outcomes were laboratory or nutritional parameters that often do not answer the real needs of patients in PC settings, where the intervention should aim to improve QoL; (2) only two treatments targeted the dyad; nevertheless, the caregiver’s involvement can be crucial when approaching the topic of food and the experiences related to it [2,15,16,17], especially in PC [21].

To date, there are no studies on the association between the psycho-educational component and the rehabilitative component of dyads to support more functional relationships for the management of CC. The psycho-educational and rehabilitative components have common objectives and modalities compared with the multimodal approaches explored in the literature. First of all, during our clinical visits, patients and their families reported attention to food preparation and mutual conflicts. We were not prepared to share communication on the management of CC, in particular, to refractory CC, and addressed the dyad, experiencing a sense of helplessness. We have detected the need for the recovery of functional skills, supporting the resources that patients and family members have, but often they cannot identify because of a lack of dedicated spaces in which to talk about it. Moreover, the need to involve the dyad in both interventions and the focus on the body, which have both approaches, determined the choice of this bimodal intervention. Finally, as SPCT with a multi-professional approach towards attention to the QoL, we preferred to follow outcomes such as proms and non-laboratories to avoid burdening patients, especially in the hospital. 

We propose a prospective mixed-methods pilot study that assesses the feasibility of a psycho-educational intervention combined with exercise among a population of cancer patients with cachexia and refractory cachexia and their caregivers, undergoing care by the SPCT [23].

## 2. Materials and Methods

The study is a mix-methods single-arm feasibility trial, conducted between March 2019 and September 2021, in S. Maria Nuova Hospital, a 900-bed public hospital located in Reggio Emilia, Italy, which was recently awarded the title of Comprehensive Clinical Cancer Center by the Organization of European Cancer Institutes (OECI).

The study was approved by the Ethics Committee Area Vasta Emilia Nord, Azienda USL-IRCCS di Reggio Emilia, Italy, number: 73/2019/SPER/ IRCCSRE. The study was conducted according to the ethical principles of the Helsinki Declaration. No changes were introduced to methods after trial commencement, and the protocol is described in detail in the protocol paper [23].

### 2.1. Aims

Primary objective: to evaluate the feasibility of psycho-educational intervention combined with the rehabilitative intervention among dyads to treat CC, assessed through completion rate.

Secondary objectives: to evaluate the QoL, caregiver burden, and physical performance;to evaluate the acceptability of the intervention among dyads.

### 2.2. Participants and Setting

This study included a convenient sample of 30 consecutive cancer patients with these characteristics: age > 18 years; good command of the Italian language; written informed consent; histologically confirmed cancer diagnosis; the presence of refractory cachexia and cachexia (assessed by the guidelines of the European Society for Clinical Nutrition and Metabolism -ESPEN3-5 guidelines [24] and the Malnutrition Universal Screening Tool (MUST) [25]); patients who have identified a caregiver. 

The exclusion criteria were as follows: the presence of an important mental disorder or dementia; severe sensory deficit; the presence of diffuse bone metastases that put the patient at risk of fracture during exercise. 

The study was conducted in SPCT, in collaboration with the Physical Medicine and Rehabilitation Unit, of Santa Maria Nuova Hospital, Azienda USL-IRCCS of Reggio Emilia.

The SPCT of the Local Health Unit—Scientific Institute for Research, Hospitalization and Healthcare (AUSL-IRCCS) hospital of Reggio Emilia is a specialized unit with no designated hospital beds. The unit includes three senior PC physicians and two advanced practice nurses. Its mission is three-pronged: patient assistance, research in PC, and specialized training to improve PC core skills in healthcare professionals.

The Physical Medicine and Rehabilitation Unit offers rehabilitation programs both for patients admitted to hospital wards and for outpatients. A specialized rehabilitation unit with 21 beds is present in the hospital, admitting patients with complex disabilities (after stroke, traumatic brain injury, complex oncological surgery, orthopedic surgery, hip fractures, etc.). The Physical Medicine and Rehabilitation Unit includes 8 physiatrists, 40 physiotherapists, 3 speech therapists, 1 occupational therapist, and 15 trained nurses. The rehabilitation team carries out clinical assistance alongside research projects in the neurological and oncological areas.

Patients were screened by SPCT, and they were informed of the diagnosis and the objectives of the therapies. Patients matching the inclusion criteria and patients who have reported awareness of the disease phase were asked to participate in the study by the PC physician.

### 2.3. Intervention

The intervention included two components: psycho-educational activities, and exercise sessions, in addition to standard care. The standard care was a specialized PC visit [26].

Figure 1 shows the timeline of the intervention and assessments. The topics addressed in the psycho-educational activities and rehabilitation sessions are described in detail in the protocol paper [24].

The nurses and the physiotherapists involved in the intervention received specific pre-enrollment training from a psychologist expert in PC and the family function family system [27].

### 2.4. The Psycho-Educational Intervention

The psycho-educational component of the intervention included 3 weekly meetings for dyads led by three trained nurses. The face-to-face consultations aimed to help the dyad cope with involuntary weight loss and declining appetite by seeking to strengthen individual and dyadic coping resources. The topics addressed included: (1) mapping changing eating habits, (2) practical examples of different ways of managing food in the care of the patients, and ways to support them, (3) re-evaluation of dyad’s needs emerged in the study period. The dyads were given an information booklet, which included a description of CC and the major emotional reactions of patients and families (see Appendix A).

The Family Relationship Index (FRI) [28] was used to collect data for the evaluation of family functioning, associating them specifically with CC. The scale was self-completed separately by the patient and the caregiver during the pre-test, and the data collected were analyzed by nurses and a psychologist (LB) before the first meeting as support in the evaluation of the dyad interaction modalities. There is a validated Italian version [28].

### 2.5. The Rehabilitative Component

The rehabilitative component of the intervention was conducted by two trained physiotherapists. It included three individual outpatient sessions in two months and three home sessions of exercises per week, carried out by the patient on his or her own or with the help of the caregiver, for a total of at least 24 home sessions + 3 outpatient face-to-face meetings with the physiotherapists over 8 weeks. The outpatient sessions with the physiotherapists focused on: (1) explaining self-management principles and goal setting; and (2) tailoring physical activity programs to help patients cope with cachexia-related fatigue, giving suggestions for the frequency, intensity, type, and timing of exercise training. During each session, the dyad was trained by the physiotherapist to perform the new tailored physical activity programme at home, including the use of weights and/or elastic bands with different strengths, if indicated.

The exercise program included strengthening exercises adapted to the patient’s clinical condition, physical performance, and preferences, as well as stretching and relaxation exercises if needed. The dyads were given an information booklet, including a list of exercises, and they were asked to keep an activity diary of their home physical activity (see Appendix A). The caregiver could help patients perform exercises or record them in the diary. There was a personal list of exercises determined in collaboration with the physiotherapist (i.e., for each exercise, there was a figure and explanation).

### 2.6. Outcomes

The completion rate was assessed for each component to evaluate the feasibility of psycho-educational intervention combined with rehabilitative intervention among dyads. Adherence to psycho-educational and physiotherapists meetings was registered and adherence to home exercise sessions was reported by patients using the activity diary.

The overall intervention was evaluated as feasible if the completion rate was ≥50% for both components.

For each participant, the following basic information was collected immediately after giving informed consent: age, sex, family unit, education, occupation, cancer diagnosis, date of diagnosis, Karnofsky performance status (KPS), body mass index (BMI), and weight loss. All replies and reasons for any refusal to participate were recorded. Data were collected anonymously and aggregated from the Azienda USL—IRCCS of Reggio Emilia database. 

To evaluate secondary objectives, the following assessments were performed:patient QoL was measured by the Functional Assessment of Anorexia-Cachexia Therapy (FAACT) [29,30],caregiver burden was measured by the Zarit Burden Scale [31,32],patient upper limb physical performance was measured by a hand-grip strength test [33],patient lower limb physical performance was measured by a 30 s sit-to-stand test [34],the acceptability of the intervention among dyads—patients and caregivers—was evaluated by ad hoc, semi-structured interviews carried out by external researchers. The FAACT and Zarit Burden Scale were self-reported questionnaires, while the hand-grip strength test and the 30-s sit-to-stand test were assessed by physiotherapists, according to the standardized scale recommendations. These evaluations aimed to collect descriptive data to establish the power calculations required for a future full-scale study. Qualitative secondary aims included the exploration of the acceptability, perceived benefits, concerns, strengths, and weaknesses of the intervention from the point of view of interviewed dyads and healthcare professionals (HPs) (see Appendix A). Ad hoc, semi-structured interviews of the dyads were conducted one month after the intervention by an external research medical doctor (GFM) with formal training and six years of experience in qualitative research in PC and by the principal investigator (LB) of the study. Both researchers did not take part in the delivery of the intervention; the participants did not know them before the interviews, and the intervention was announced by the recruiters (PC professionals during ordinary care).

Nurses and physiotherapists who participated in the study were interviewed in a single focus group session by an external research medical doctor (GFM) at the end of the study to explore HPs’ points of view on the bimodal intervention and dyads’ adherence and coping with the program (see Appendix A).

### 2.7. Sample Size and Statistical Analysis

In absence of an a priori hypothesis, given the exploratory nature of the study, no formal sample size calculation was performed; the research team planned to include a sample of 30 consecutive cancer patients with cachexia and refractory cachexia, addressed to SPCT. Clinical and demographic data were expressed in terms of frequency and percentage for categorical variables and the mean ± SD for quantitative variables. The percentage of compliance with the treatment (the main study focus) was accompanied by a 95% Clopper-Pearson two-tailed confidence interval. Secondary outcomes/scores were descriptively summarized. Statistical analysis was performed using R 4.0.4.

### 2.8. Qualitative Analysis

Recordings of the interviews were transcribed verbatim and then analyzed using thematic analysis to explore the content and context of the responses [35,36]. The qualitative evaluation was performed and reported by the consolidated criteria for reporting qualitative research (COREQ) guidelines [37]. Two researchers (LB and GFM) independently labeled each transcript, resolving differences in labeling. Throughout an iterative process, they inductively identified a few sub-themes. Finally, they revised the preliminary thematic analysis and regrouped and renamed some themes and sub-themes to get to a unified and agreed version. This revision was then discussed and amended with the other HPs involved in the study.

Qualitative coding was performed using colors and comments in Microsoft Word, as no specific qualitative analysis software was needed due to the manageable quantity of materials. The inductively built coding tree resulting from both researchers’ analysis followed the structure of the intervention and, in part, the structure of the interview (comments on psycho-social intervention + physiotherapist intervention + comment on inter-professional interaction and possible suggestions for future developments) and so it was agreed to present the data in this form. Even though it is generally preferred not to keep the interview’s structure in qualitative data presentation, we chose this disposition of data as most useful for the understanding of a piloting experience. 

## 3. Results

Patient enrollment started in May 2019 and concluded in September 2021. Due to the COVID-19 pandemic, the overall duration of the study was extended from 24 months to 29 months because enrollment was interrupted for six months during lockdown periods. Moreover, the study was closed before the completion of recruitment in September 2021 because the planned interval for conducting it had exceeded the duration of the physiotherapists’ employment contract, supported by public funding (Bando per la valorizzazione della Ricerca Istituzionale in ambito oncologico 2018—Fondi per Mille 2016). 

Twenty-four dyads corresponding to the inclusion criteria consented to participate. 

Seven patients refused to attend because the number of meetings was too demanding; reporting that they had been to the hospital for many visits. Forty-seven patients were not considered eligible for the following reasons: poor prognosis (*N* = 30), absence/unavailability of a caregiver (*N* = 9), lack of knowledge of the Italian language (*N* = 4), and other reasons (*N* = 4) (presence of mental disorder, non-awareness of diagnosis). Figure 2 shows the flow diagram of the study. 

### 3.1. Socio-Demographic and Clinical Data

The socio-demographic and clinical characteristics of the participants are reported in Table 1. 

The patients’ mean age was 66 years old, and 62.5% of enrolled patients were male. Almost all patients lived with their families (92%). All patients had a good performance status (KPS ≥ 70). Eighty-three percent of the sample presented cachexia, twenty-five percent of patients were affected by pancreatic cancer, and twenty-one percent of them had lung cancer. People were enrolled at a mean time of 2 years from a cancer diagnosis.

### 3.2. Feasibility Data

Twenty-four dyads were evaluated at baseline (T0), sixteen at T1, 11 at T2, and six at T3 final follow-up.

Twenty dyads received at least two psycho-educational sessions, so the psycho-educational component was feasible for 83.3% of the sample. 

Six dyads participated in at least fourteen out of twenty-seven exercise sessions, so the rehabilitative component was feasible for 25.0% of the sample. Six dyads showed compliance greater than 50% for both components of the bimodal intervention. Table 2 shows adherence to the intervention.

Eighteen patients withdrew from the study, and the causes of withdrawal were: death (*n* = 1), deterioration of clinical conditions (*n* = 16), and difficulties coming back for follow-up (*n* = 11). One patient was affected by a pathological fracture, but this did not occur during exercise. Twelve out of twenty-four patients (50%) died within 3 months of enrollment (see Table 1).

### 3.3. Secondary Outcomes

The changes over time in secondary outcomes are reported in Table 3. 

QoL decreased over time, and the caregiver burden diminished between enrollment and T2.

Upper limb strength was substantially stable in the first month (between T0 and T2) and worsened at T3. Lower limb physical performance measured by the 30-s sit-to-stand test showed better scores at T3.

Table 3 includes the scores of all the evaluations made on patients and caregivers at the time points scheduled in the protocol. Table 4 includes only the six patients and their caregivers that were evaluated at every assessment point, from T0 to T3, and who were compliant with the overall program.

If we consider the scores of patients evaluated at 2 months follow-up (T3), the trend shows no deterioration in QoL, caregiver burden, or patients’ physical performance (see Table 4). 

### 3.4. Qualitative Analysis

Six dyads who completed the intervention accepted to be interviewed. Interviews took place within the hospital and lasted between 30 and 50 min. All HPs (three PC nurses and two physiotherapists) involved in the study accepted to be interviewed in a single focus group session, which lasted one h. 

Five macro themes/categories and some suggestions were identified: (1) bimodal intervention, (2) psycho-educational intervention, (3) physiotherapist intervention, (4) inter-professional collaboration, and (5) dyad as the intervention’s target. 

The following paragraphs describe the themes from the point of view of the dyads and the HPs. Table 5 lists the macro themes/categories, sub-themes, and representative quotations from the participants’ interviews.

#### 3.4.1. Bimodal Intervention

From the point of view of the dyads, the bimodal intervention received great appreciation. For example, they reported a positive perception of being engaged in a non-clinical intervention and having the possibility to go to the hospital also to talk about the food and their relationship, not only to do the therapies or exams: “*Having a particular commitment different from the usual (…), not to be done necessarily like therapies”* (D3, patient).

They reported a positive perception of the proposed intervention by the PC physicians because they referred to a relationship of trust. Moreover, they appreciated the “*feeling*” part of the care process, which improved care for other patients as well: “*I like to participate in these projects because in my heart I feel like helping (…)*” (D1, patient). Finally, they reported a positive perception of organizing the schedule, as acceptability of the number and frequency of meetings on the same day. 

From the point of view of HPs, being part of two different teams and two different departments made coordination harder. This issue also emerged in regards to the use of the PC language, favored over time by supervisors: “*Before COVID (…) we had the chance to meet more often with the nurses, and this was useful as we were more able to tailor the intervention on patient’s needs*” (FG, I1, physiotherapist). This was an opportunity to improve the training for the physiotherapists in PC. Particularly to the dyads, HPs also appreciated doing the intervention on the same day as scheduled appointments for other treatments, but it was highly stressful due to time constraints: “*If you have a high number of patients, it might be possible to schedule fixed timetables for this kind of patients”* (FG, I3, nurse) or else. But trying to “squeeze” these appointments into the routine care of patients guaranteed a higher adherence to the intervention. 

#### 3.4.2. Psycho-Educational Intervention

The dyads appreciated having the chance to talk about their nutrition in a dedicated space and time with a specialist. They reported a positive perception of the validation of their desires and nutritional habits, as listening to the needs of the patient and caregiver with a non-judgmental approach, proposing open questions to favor the understanding of the different points of view of the patient and the caregiver: “*Then slowly with the nurse who asked you the questions, asked you, they told you, he vented… it helped us so much*” (D2, patient). It was important to talk about CC and to have specific information, supported by a booklet, in contrast with internet/friends (sometimes misleading) opinions. The dyads reported that the reflections and suggestions received had a positive impact on the relationship about food that continued over time: “*I continued to follow the advice they gave me, it was very useful, even after, and I kept talking about it with my daughter. She still asks me ‘Did you do this exercise?*” (D4, patient). Finally, the dyad knew that the PC nurse worked with the PC physician, reporting the perception of being assisted in a global approach and continuity with the doctor.

HPs reported some difficulties with the new role of psycho-education on CC; for example, some dyads confused them with a dietitian or nutritionist. They express the need to become familiar with a psycho-educational intervention because they provide psychological support that is integrated with clinical information. They perceived the results of the intervention at the end of the three meetings, particularly with the dyad that reports understanding the sense of the psycho-education on CC over time: “*The impact of these interventions is not immediate. Dyads started a bit perplexed about the psycho-social intervention, but then it became a space to share their thoughts. It became an appointment they didn’t want to miss: they never missed them*” (FG, I3, nurse). They experience positive expectations due to the relevant role of the nurse in a psycho-educational intervention, and enrollment was easier because of the presence of the PT intervention, which was perceived as a value by the dyad.

#### 3.4.3. Physiotherapist Intervention

The dyad reported generally appreciating the intervention. In particular, they enhance the personalization of the intervention: “*I tried to adapt based on how I was, the pain, how I felt, as the PT told me*” (D3, patient). Particularly, some patients needed to interrupt or modify the intervention due to disease or treatment consequences.

For HPs working with late-stage patients was perceived as really hard: “*Working patients conditions could worsen fast, and the intervention that we thought to carry on quickly became too demanding for him (…) in some patients I felt like I was out of my comfort zone*” (FG I2, physiotherapist). Particularly, they appreciated the supervision to construct a common language of PC. Finally, they reflected on the selection of the hand-grip and sit-stand tests that were able to measure the performance but not the lifestyle modification: “*The big difference of a lifestyle intervention might not be assessed with these tests like “hand-grip”, maybe other tests, like the EORTC, might be more appropriate*” (FG I2, physiotherapist).

#### 3.4.4. Inter-Professional Collaboration

No significant organizational problem or suggestion emerged from the dyad about the integration of the two interventions. 

HPs reported that periodic meetings, when possible, were a facilitator factor for inter-professional collaboration: made it easier to share background information on the dyad and clarify the treatment’s goals: “*Better understanding of (…) patient comprehension of their disease helped our intervention (…)*” (FG, I1, physiotherapist). 

#### 3.4.5. Dyad as the Intervention’s Target

The patient perceived the caregiver as motivational: “*I’ve been involved, I’ve always come, at home I look at her when she does the exercises, we talk about it*” (D1, caregiver). In general, the caregiver felt that he/she was helping the patient.

Particularly, HPs observed that family members were often motivated and encouraging to patients: “*Facing some difficult topics with the presence of a family member often was helpful for the patient. Caregivers were often the ones who made more questions*” (FG, I2, physiotherapist). 

#### 3.4.6. Suggestions

The final part of the interview was dedicated to the relief of suggestions for improving the project. The dyads reported that group activities, such as physiotherapist activities and cooking activities with other dyads or patients, might be motivating. 

The HPs sustained that periodic meetings focused on a shared dyad to coordinate the intervention. They also suggested adding flexibility in the delivery of physiotherapist interventions: “*Some patients may be fitter and with a more helping caregiver, might be seen every two weeks, while others might need a closer to follow-up to get all the benefits of such an intervention*” (FG, I2, physiotherapist).

## 4. Discussion

Our study evaluated the feasibility and acceptability of a psycho-educational intervention combined with exercise among a population of cancer patients with cachexia and refractory cachexia and their caregivers, undergoing care by the SPCT [24], both from the patient’s and HP’s perspectives. Our findings strongly support the acceptability of the intervention, as evaluated by a semi-structured interview with dyads, but only partially support its feasibility. We present two potential reasons why this trial did not meet all feasibility goals. 

One potential factor affecting feasibility is the unpredictable course of cancer malignancies. Only six dyads showed compliance greater than 50% for both components of the bimodal intervention. In particular, the psycho-educational component was feasible for 83.3% of the sample, while the rehabilitative component was feasible for 25.0% of the sample. The most frequent reason for drop-out was the deterioration of clinical conditions, which prevented patients from completing the exercise program and the 2-month follow-up. On the other side, the 6 patients who did not experience a rapid clinical condition deterioration were compliant with the overall intervention, both in the psycho-educational and the rehabilitative components. Twelve out of twenty-four patients (50%) died within 3 months of enrollment. These data are in line with other studies conducted on advanced patients with CC [14,15]. The PC physicians who enrolled failed to provide the right prognosis evaluation, as witnessed by the high rate of dropout during the 3 months of the rehabilitative part. Moreover, we believe that dyads more easily completed psycho-educational intervention because it ended in a shorter time (three weeks), unlike rehabilitation, which required a two-month follow-up. The exercise program was built taking into account the best evidence available in 2018 and focused mainly on resistance exercises and goal setting, including several home-based sessions with the help of patients’ caregivers if needed, but probably was more demanding than talk in the educational component for cachectic patients. The duration of the exercise intervention over 8 weeks was similar to exercise programs proposed by other authors [38], and it had a clinical rationale to be effective on muscle mass and strength. In the six patients evaluated at two months follow-up, the hand-grip and sit-to-stand tests were stable, and muscle strength did not wane. In these patients, the repetition of resistance exercise three times per week associated with flexibility exercises avoided deterioration of upper and lower limb physical performances. Maybe a future study could explore a different modality to complete supervised exercise in patients with advanced cancer after the first month from enrollment, such as a physiotherapist’s phone call or telerehabilitation [39]. This could support the most fragile people by avoiding the demanding and tiring journey to the hospital.

The second factor is staffing. We trained three nurses and two physiotherapists intending to enroll 30 dyads. The time for enrollment was longer than expected because the pandemic had a great impact on cancer patients’ access to hospital care. The physiotherapists left the practice before completing the planned enrollment of 30 dyads, but they completed the intervention and the evaluations for all 24 dyads included in the study. We did not have the resources to train other physiotherapists. Consequently, we enrolled 24 out of 30 planned dyads. Our experience suggests conducting studies with HPs who work within the teams and who are not recruited only for specific research. This could help implement the assistance that has been learned from the research study.

The strengths of the study were several. 

Qualitative data showed the acceptability of the bimodal intervention through semistructured interviews with the dyads. The method followed for qualitative assessment was the Framework Method, which is particularly suitable for helping an inter-professional and interdisciplinary research team analyze and manage qualitative data. Researchers use this method to address research questions concerning the meaning people give to their experience, allowing a comprehensive thematic analysis that can be shared and discussed within the research team. Its main feature is to provide the researchers with a matrix that is used to analyze the data and highlight differences and/or commonalities within the resulting themes [40]. 

First of all, we proposed a psycho-educational intervention with the nurses within the SPCT to avoid some criticism in other similar studies [41,42,43], in which it was unclear if the nurses worked with a medical team and if they could activate other HPs to support communication about the clinical condition. Particularly, the dyads appreciated that the PC nurse worked with the PC physician, reporting the perception of being assisted in a global approach and continuity with the doctor, as emerged from the qualitative data. 

In addition, qualitative data showed the utility of psycho-educational intervention for the dyads, suggesting that they need honest and problem-centered communication, as reported in other studies [16,19]. A large survey (76% response rate) of 702 bereaved family members of cancer patients in Japan showed that those who believed they forced the patient to eat to avoid death and those who believed they did not have correct information about CC showed a higher risk of bereavement depression [20]. HPs should empower the dyad to understand the refractory and progressive nature of cachexia and acknowledge its effects.

Regarding the rehabilitation intervention, we can say that the program centered on home exercise facilitated adherence for patients who did not experience rapid deterioration in their clinical conditions and reached a 2-month follow-up. Other authors [44] suggested offering people with advanced cancer home-based rehabilitation programs, to maximize compliance, but including supervised exercise may enhance the effectiveness [45]. The individualized exercise prescription by a trained physiotherapist is in line with ESMO CC recommendations [2].

Finally, HPs found that periodic meetings were a facilitator factor for inter-professional collaboration. Working with late-stage patients was perceived as hard for the physiotherapists, and the supervision supported them to construct a common language of PC.

Future PC intervention trials for CC will be informed by several insights gained from this pilot study. First, planning for staff attrition and securing protected time for HP’s interventionists are crucial for meeting recruitment targets and study timelines and implementing the intervention. Second, we recommend short bimodal interventions over a maximum of one month with a weekly meeting, supporting the personalized exercises for the patient at home. Similar trials should consider focusing on identifying patients with CC needs early during hospitalization and assessing the prognosis with specific instruments (e.g., the Palliative Prognostic Index [46]). The interventions should be flexible in frequency and duration to facilitate the participation of dyads. Finally, we suggested identifying questionnaires that are more focused on PC needs and QoL, especially for the physiotherapist intervention.

## 5. Conclusions

In summary, we describe the feasibility and acceptability findings from a pilot study of a bimodal intervention for CC that addressed the dyad. While we did not meet our primary feasibility endpoint and had mixed acceptability, our experience provides insight into the challenges and lessons learned in implementing a primary PC intervention in this population.

More robust studies are needed to help clinicians understand the best exercise program for CC patients, to be included in a multimodal intervention.

## Figures and Tables

**Figure 1 cancers-15-02063-f001:**
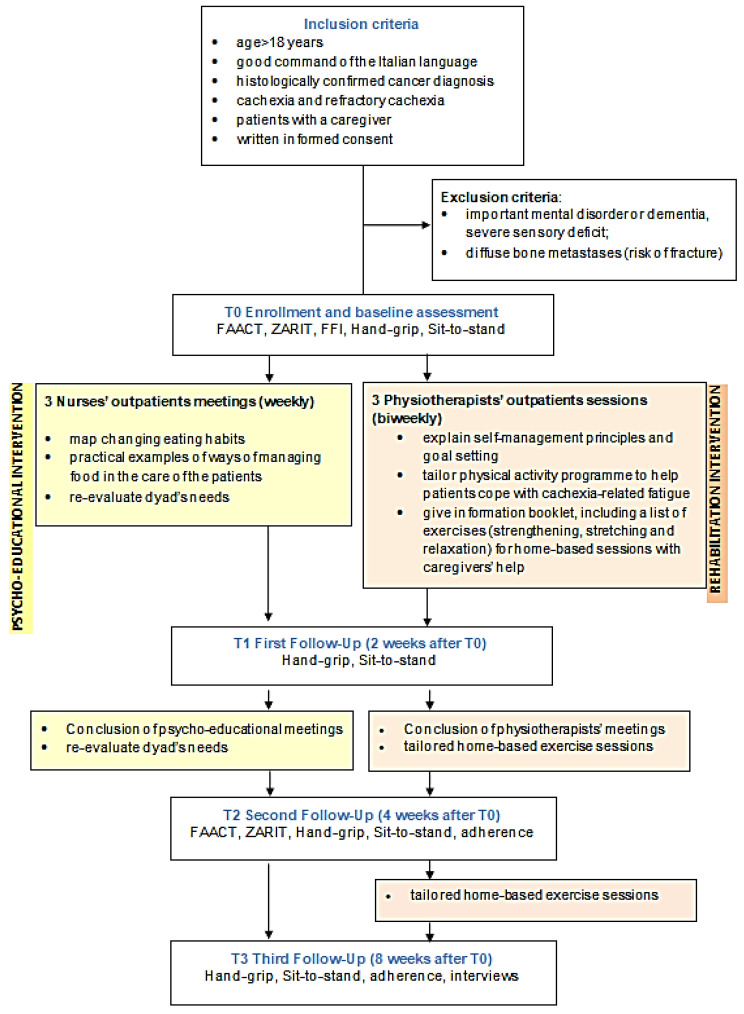
Timeline of assessments and interventions.

**Figure 2 cancers-15-02063-f002:**
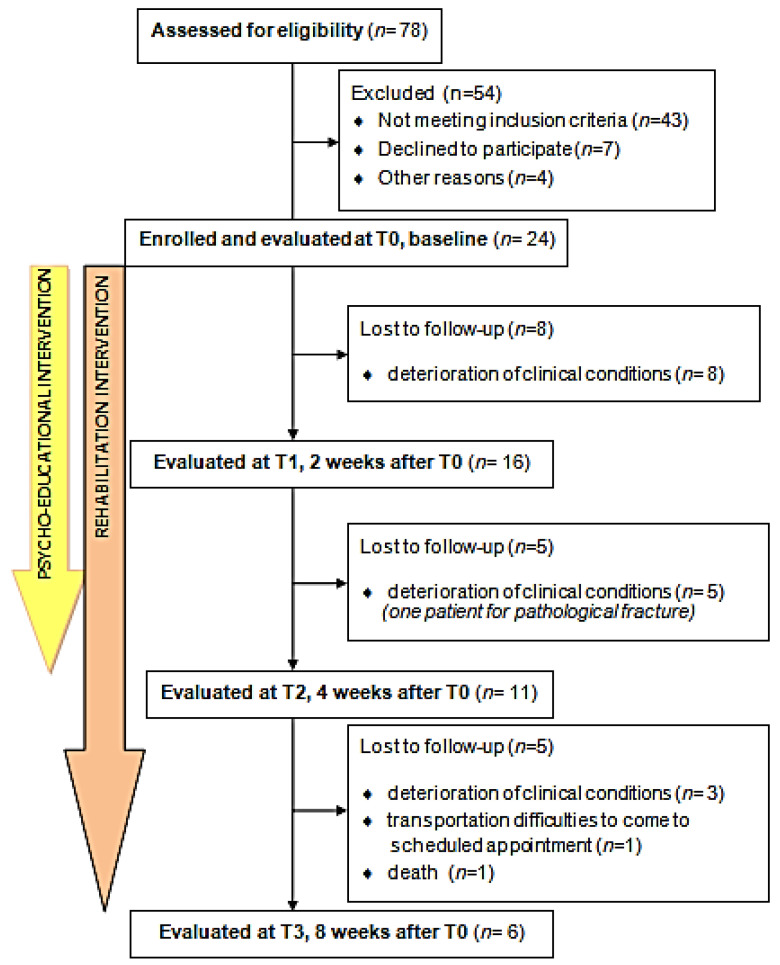
Flow diagram of enrollment.

**Table 1 cancers-15-02063-t001:** Patients sociodemographic and clinical characteristics.

Characteristics	Variables	*N* = 24
Age	Mean, years (±SD)	65.96 (±10.61)
Sex	M (*n*,%)	15 (62.5%)
	F (*n*,%)	9 (37.5%)
Education	P Primary school (*n*,%)	2 (9.5%)
	Secondary school (*n*,%)	6 (28.6%)
	High school (*n*,%)	11 (55.4%)
	Bachelor’s degree (*n*,%)	2 (9.5%)
Occupation	Employed (*n*,%)	9 (37.5%)
	Retired (*n*,%)	13 (54.2%)
	Other (*n*,%)	2 (8.4%)
Family unit	Live alone (*n*,%)	2 (8.4%)
	Live with others (*n*,%)	22 (91.7%)
Diagnosis	Pancreatic cancer (*n*,%)	6 (25.0%)
	Lung cancer (*n*,%)	5 (20.8%)
	Renal cancer (*n*,%)	3 (12.5%)
	Upper GI cancer (*n*,%)	3 (12.5%)
	Bladder cancer (*n*,%)	2 (8.3%)
	Other (*n*,%)	5 (20.8%)
Time from diagnosis at enrollment	Years, *mean* (±*SD*)	2.42 (±4.90)
Karnofsky Performance Status	70 (*n*,%)	4 (17.4%)
	80 (*n*,%)	13 (56.5%)
	90 (*n*,%)	6 (26.1%)
Cachexia	Reversible (*n*,%)	20 (83.3%)
	Refractory cachexia (*n*,%)	4 (16.7%)
Body Mass Index	Usual value, *mean* (±*SD*)	26.68 (±4.74)
	Current value, *mean* (±*SD*)	22.12 (±4.41)
Weight loss	Usual value, *mean kg* (±*SD*)	79.17 (±18.11)
	Current value, *mean kg* (±*SD*)	66.00 (±15.80)
Withdrawals from the study	Deterioration of clinical conditions (*n*, %)	16 (66.7%)
	Death (*n*, %)	1 (4.2%)
	Difficulties to come to follow-up (transportation) (*n*,%)	1 (4.2%)
N of patients dead within 3 months from enrollment	(*n*,%)	12 (50.0%)

**Table 2 cancers-15-02063-t002:** Patient compliance with intervention.

Treatment	Compliant (*N* out of 24)	Percentage (95% CI)
Psycho-educational	20	83.3 (62.6–95.3)
Rehabilitation	6	25.0 (9.8–46.7)
Overall	6	25.0 (9.8–46.7)

**Table 3 cancers-15-02063-t003:** Secondary outcomes: changes over time.

Assessment	T0 Mean (±SD)	T1 Mean (±SD)	T2 Mean (±SD)	T3 Mean (±SD)	Difference T2–T0 Mean (±SD)
FAACT score (patients QoL)	(*n* = 24) 18.42 (± 5.81)		(*n* = 6) 15.17 (± 2.93)		(*n* = 6) 1.00 (± 4.94)
ZARIT score (caregivers burden)	(*n* = 24) 24.08 (± 12.95)		(*n* = 6) 18.33 (± 9.44)		(*n* = 6) 0.50 (± 6.47)
Hand-grip (patients upper limbs performance)	(*n* = 23) 23.96 (± 9.57)	(*n* = 16) 23.16 (± 7.12)	(*n* = 10) 24.44 (± 7.67)	(*n* = 5) 20.69 (± 10.57)	(*n* = 10) −0.37 (± 2.53)
Sit-to-stand (N of repetition in 30 s) (patients lower limbs performance)	(*n* = 23) 8.13 (± 3.66)	(*n* = 14) 7.79 (± 2.39)	(*n* = 10) 7.90 (± 4.82)	(*n* = 5) 9.80 (± 2.28)	(*n* = 10) −0.70 (± 3.06)

**Table 4 cancers-15-02063-t004:** Secondary outcomes: changes over time in patients compliant with the overall program.

Assessment	T0 Mean (±SD)	T1 Mean (±SD)	T2 Mean (±SD)	T3 Mean (±SD)
FAACT score (patients QoL)	(*n* = 6) 15.50 (± 4.32)		(*n* = 6) 15.17 (± 2.93)	
ZARIT score (caregivers burden)	(*n* = 6) 17.83 (± 13.26)		(*n* = 4) 17.50 (± 11.96)	
Hand-grip (patients upper limbs performance)	(*n* = 6) 21.90 (± 8.77)	(*n* = 6) 21.97 (± 6.99)	(*n* = 6) 21.29 (± 8.50)	(*n* = 5) 20.69 (± 10.57)
Sit-to-stand (N of repetition in 30 s) (patients lower limbs performance)	(*n* = 6) 9.17 (± 1.47)	(*n* = 6) 8.50 (± 1.05)	(*n* = 6) 9.50 (± 2.51)	(*n* = 5) 9.80 (± 2.28)

**Table 5 cancers-15-02063-t005:** Qualitative data aspects of the intervention according to the dyads’ and healthcare professionals’ experiences, and representative quotations.

Stages or Aspects of the Intervention	Dyads (Patient + Caregiver) *N* = 6	HPs (3 PC Nurse + 2 PT) *N* = 5
Bimodal intervention	- Positive perception of being engaged in a non-clinical intervention “*Having a particular commitment different from the usual, because I took it just as a pleasant, welcome commitment, not to be done necessarily like therapies or visits or coming to the hospital because you are sick*” (D3, patient) “*We wanted to help our father with food, so we were very happy*” (D3, caregiver) - Positive perception of the proposal of the intervention by PC physicians (relationship of trust) “*My physician told me about the project, knowing my character and how she had seen me, tired and worried. After that my daughter also came and we talked about all three. It’s been useful*” (D4, patient) “*My physician told me about the project, then my husband also came and we talked about it together, because I was afraid to weigh on my husband. She helped us*” (D2, patient) - Positive perception of the research theme (feeling part of the care process, improving care for other patients as well) *“I like to participate in these projects because in my heart I feel like helping, I like to give a hand to other patients”* (D1, patient) *“I thought that there was research and that I agreed, at least every patient if he confesses his pain can favor others. I agreed from the beginning. Helping the community”* (D3, patient) - Positive perception of organizing the schedule (different meetings on the same day) *“With the times and the organization it went very well, because I never had to come to Reggio on purpose; they have always tried to put these meetings together with others”* (D1, patient) “*It was not challenging with the therapies, because the nursed changed the days according to the therapies, I’m really good, there has never been a problem*” (D2, patient)	- Having the intervention during the same day as scheduled appointments for other treatments was appreciated but highly stressful due to time constraints “*Before COVID (…) we had the chance to meet more often with the nurses, and this was useful as we were more able to tailor the intervention on patient’s needs* ” (FG, I1, physiotherapist) *“If you have a high number of patients, it might be possible to schedule fixed timetables for this kind of patients”* (FG, I3, nurse) - Need to construct a common language on PC favored over time by supervisions “*If it’s been more demanding for us*” (FG, I2, physiotherapist) *“Having periodic meetings helped (…) as they have a different point of view (…) and might have a deeper knowledge of the patients*” (FG, I2, physiotherapist) “*In some cases we had the chance to exchange opinions on patients and caregivers more, and this made a difference”* (FG, I3, nurse)
Psychoeducational intervention	- Appreciated having the possibility to talk about CC *“At the beginning it was my husband more nervous, maybe he could not get into the order of ideas of what I had, how I was, then slowly with the nurse who asked you the questions, asked you, they told you, he vented… it helped us so much”* (D2, patient) *“Certain things that were negative to me, I realized that in the conditions in which she was, they were not”* (D2, caregiver) *“It was the first time we talked so thoroughly about food. Before we talked about it in a medical way, but this intervention is not everywhere, it is not an aspect to be underestimated”* (D3, caregiver) Appreciation of the information about CC (booklet) “*I didn’t even know what cachexia was”* (D2, patient) *“I even went to the dictionary to look, otherwise you remain ignorant*” (D2, caregiver) “*Having instructions on what not to do to facilitate the work of clinicians was fundamental. So knowing what to do so as not to ruin a path is very important*” (D3, caregiver) *“Have personalized information, because maybe you hear someone who has had the same experience but tells you wrong things; you feel abandoned, even talking about it right away, it’s important”* (D5, caregiver) Positive impact on the relationship about food that continues over time *“Many times we do not eat the same things; I continue to cook for him, while I had to change my diet, so we make two different menus. But it doesn’t bother us”* (D1, patient) *“At the beginning you stayed a bit like that, what is it for… it’s a new thing, but with the passage of time it has helped us a lot actually”* (D2, caregiver) *“I continued to follow the advice they gave me, it was very useful, even after, and I kept talking about it with my daughter. She still asks me ‘Did you do this exercise’”* (D4, patient) Positive perception of the nursing role within PC *“We already knew the nurse, so we were pleased that she was all inside PC* (D3) *“She told us that some things she would talk about it with the physician, pleased us we felt followed”* (D5, caregiver)	- Initial confusion with a dietitian/nutritionist’s intervention *“At the beginning, it was not very clear to them. It became clearer going on with the first intervention” (FG, I3, nurse)* *“We had to explain more times that we weren’t dieticians, and clarify the real objectives of the meetings”* (FG, I4, nurse) - Perception of results at the end of the three meetings *“The impact of these interventions is not immediate. Dyads started a bit perplexed about the psychosocial intervention, but then it became a space to share their thoughts. It became an appointment they didn’t really want to miss: they never missed them”* (FG, I3, nurse) - Enrollment was easier because of the presence of the PT intervention, perceived as a value from the dyad *“In the beginning, the idea of a physiotherapy intervention might have helped. The enrollments (…) then they became increasingly motivated”* (FG, I5, nurse)
PT intervention	- Positive perception of the personalization of the intervention *"It is very complete as a physical path, it invests all the muscles, not just the legs or arms, it is complete, I felt more harmonious as a body*" (D1, patient) *“I tried to adapt based on how I was, the pain, how I felt, as the PT told me”* (D3, patient) *“It is something that serves, it is the right thing that helps him not to lose calories, we always do the exercises according to the days”* (D3, caregiver) *“She explained to me the various reasons why I have to do the exercises. The various exercises he gave me he told me to do even more, and I did them, but without getting tired”* (D4, patient) - Often needed to interrupt or module the intervention due to disease or treatments consequences *“I had to lighten the exercises that concern the arms, because I have the port, in my opinion I was not even asked if I have the port, but it affects. So I didn’t eliminate the beings for the arms, I decreased both in number and intensity”* (D1, patient) *“These last few weeks, I gave up gymnastics because I already had so many problems”* (D2, patient)	- Working with late-stage patients was perceived as really hard *“Working patients conditions could worsen really fast, and the intervention that we thought to carry on quickly became too demanding for him (…) in some patients I felt like I was out of my comfort zone”* (FG, I2, physiotherapist) - Hand-grip and sit-stand tests were able to measure the performance but not the lifestyle modification *“The big difference of a lifestyle intervention might not be assessed with these tests like “hand-grip”, maybe other tests, like the EORTC, might be more appropriate”* (FG, I2, physiotherapist)
Inter-professional collaboration	No specific difficulties perceived	- Periodic meetings, when possible, were a facilitator factor: made it easier to share background information on D and clarify the treatment’s goals “*If we had less chances to coordinate care, it showed! (…) When we had the chance to meet with the whole team it was a facilitator for a better care”* (FG, I4, nurse) *“Better understanding of (…) patient comprehension of their disease helped our intervention”* (…) (FG, I1, physiotherapist) “*They (the nurses delivering the psychosocial intervention) might have a better understanding of their hopes, and periodic meetings helped us treating them accordingly”* (FG, I1, physiotherapist)”
Dyad as the intervention’s target	- The caregiver was perceived as a motivational aspect by the patient “I’ve been involved, I’ve always come, at home I look at her when she does the exercises, we talk about it” (D1, caregiver) - The caregiver felt that he/she was helping the patient *“We wanted to help him but I didn’t know how to do it because we were afraid of doing wrong things. being able to understand together what to do was useful, for us and for our father”* (D3, caregiver) *“For my daughter it was useful to see how they helped me in eating and doing the exercises, seeing me engaged”* (D4, patient)	- Family members were often motivated and encouraging to patients *“Facing some difficult topics with the presence of a family member often was helpful for the patient. Caregivers were often the ones who made more questions”* (FG, I2, physiotherapist) “*On some problems [the caregivers] helped us (…) to understand some problems that were emerging during the day and which could be the right strategies to better cope with them”* (FG, I4, nurse)
Suggestions	- *Group activities might be motivating* *"It would be nice in such a project to do gymnastics together with some other patient, small groups, there would be a comparison, a help to each other, even just a few meetings"* (D1, patient) *"Make groups for example on how to cook, do them together as they do for example in other centers”* (D5, patient)	- Periodic meeting focused on shared D to coordinate the intervention “*Some patients may be more fit and with a more helping caregiver, might be seen every two weeks, while others might need a closer to follow-up to get all the benefits of such an intervention”* (FG, I2, physiotherapist)

## Data Availability

The data that support the findings of this study are available from the first author and the corresponding author [L.B., S.F.], upon reasonable request.

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
