# Peer review of "Psycho-Educational and Rehabilitative Intervention to Manage Cancer Cachexia (PRICC) for Advanced Patients and Their Caregivers: Lessons Learned from a Single-Arm Feasibility Trial"

_cancers, 2023, doi:10.3390/cancers15072063_

Round 1

Reviewer 1 Report

This is an interesting and important pilot study in supportive care in cancer. Publishing of the study will provide contribution to clinical literature in this area.

Comments:

Paper does not explain which measures were completed by patients and which by caregivers. Are tables and figures reporting data for patients or caregivers? Are any caregiver data reported in this paper?

PROMS should be spelled out (Simple Summary)

Page 6 Outcomes "To evaluate secondary objectives..." For each outcome specify if it is patients or caregivers.

In statistical analysis section, explain how IQR was calculated.

Figure 1: Figure appears blurry and there seem to be spelling errors.

Figure 2: Add time point/level that will correspond to completing at least two sessions (n=20)

Table 1: Consider spelling out acronyms (e.g., KPS). Consider consistent capitalization.

Table 4: Numbers in Tables 4 are not consistent with numbers in Table 3. For example, FAACT: T0=15.5 T2=15.5 giving diff=0, however Table 3 states diff=1.0

Author Response

We thank reviewer 1 for all the comments.

Please see the attachment for the point-by point response (we include the response to all the comments made by the two reviewers, to allow each of them to find the changes in the Manuscript easily). 

Reviewer 2 Report

This manuscript reports on an exploratory one-armed trial to monitor compliance with a combined intervention of psycho-education and exercise training in patients with cancer cachexia and a corresponding caregiver. 80% of invited dyads participated, however 75% of these withdrew during the trial; the main reason for withdrawal was worsening of the clinical condition or even death. These data are interesting and, though somewhat disappointing, worth reporting.

I have the following comments:

1. The terms ‘exercise’ and ‘rehabilitation’ are used interchangeably; however, in some countries, ‘rehabilitation’ may include more interventions than only exercise training. Therefore, I recommend to here only use the term ‘exercise’.

2. In the Material and Methods section, line #5 from the bottom, the authors refer to ‘irreversible cachexia’. This is an unusual term, please give a reference and definition. The Consensus definition of 2011 (Fearon et al.) presented ‘refractory’ cachexia, but included in the definition an expected survival of <3 months. This would collide with the exclusion on page 3, line #2 from bottom, of ‘survival < 3 months’. This would be a contradiction in design.

3. In the Material and Methods section, line #4 from the bottom, the authors refer to ‘ESPEN3-5 guidelines’ and ‘MUST’ calculation. Please be more specific what this is meant to refer to: different ESPEN guidelines? Please give references. MUST screening? Screening is no diagnosis and should not be referred to here, unless in a different way.

4. In addition, in the Material and Methods section, line #2 from the bottom, the authors refer to an exclusion criterion of expected prognosis < 3 months of survival. However, 50% of included patients died during the 3-month trial (page 9): Was the exclusion criterion known to the palliative care team responsible for including patients?

5. Table 1 does not include data for reversible or irreversible cachexia.

6. Page 11, line #1 and line #3: who ‘accepted’: did dyads and HPs accept interviews or did the authors accept dyads and HPs to be interviewed? Please describe more clearly. Maybe instead of ‘were accepted’ it should read ‘accepted’?

7. Discussion, page 13, line #4: What is meant by the term ‘acceptability’? The fact that 24 out of 30 dyads accepted participation? To have a drop-out rate of 75% appears to indicate lower ‘acceptability’.

8. Discussion, paragraph 3, line #4: The authors state that the physiotherapists left before completing any study visits. Is this true such that no study visits took place? What was a ‘study visit’, an exercise intervention?

9. Discussion, page 14, paragraph #3: The authors should not favor short interventions (second recommendation) just because there was a high drop-out rate. Shorter trials might be even less effective. Instead future trials should focus on better observing exclusion criteria to ensure inclusion of patients who may benefit from the study interventions.

Author Response

We thank reviewer 2 for all the comments.

Please see the attachment for the point-by point response (we include the response to all the comments made by the two reviewers, to allow each of them to find the changes in the Manuscript easily). 

Round 2

Reviewer 1 Report

Authors made changes that improved the quality of the manuscript.

However, there is still some disagreement in numbers between table 3 & 4. For example, FAACT at time T2 in both Table 3 and Table 4 there is stated n=6 patients, what suggests that these are the same patients. However, mean values are different 15.17 vs 15.50.  This reviewer suggests that authors carefully recheck all numbers in Table 3 & Table 4.

Author Response

Thank you so much. We have checked again Table 3 and Table 4 and you are right, there was a typo in FAACT score at T2 evaluation. The six patients were the same, and the correct mean value is 15.17. We have changed it in Table 4.

Round 3

Reviewer 1 Report

Author addressed the items brought by this reviewer.